# Long term monitoring of the geoelectric field in the UK – 2012-2024

Robert Lyon<sup>1</sup>, Gemma S. Richardson<sup>1</sup>, Orsi Baillie<sup>1</sup>

<sup>1</sup>British Geological Survey, Research Ave South, Riccarton, EH14 4AP Edinburgh, UK

Correspondence to: Robert Lyon (rlyon@bgs.ac.uk)

Abstract. During severe geomagnetic storms, rapid changes in the Earth's magnetic field can induce a significant geoelectric field in the conductive subsurface. This field can cause large potential differences in grounded systems that have long conductors between earthing points, such as power lines, and cause currents to flow between them. These currents are known as Geomagnetically Induced Currents (GIC). To predict the effects on ground-based infrastructure from geomagnetic storms during space weather events, estimates of the subsurface electrical resistivity are required. These can be constructed using transfer functions between the magnetic and electric field, calculated from measurements made at short-term monitoring installations, recording for days to weeks. However, longer-term monitoring of years to decades is valuable too, as this provides a large and rich set of data encompassing both quiet periods and storms that can be used to enhance and ground-truth geoelectric field and GIC estimates. There are a limited number of permanent monitoring systems around the world, and until 2012 there were none in the UK aside from historical measurements made in the 19th century. The British Geological Survey (BGS) installed three geoelectric field monitoring sites in 2012 and 2013 co-located with our INTERMAGNET observatories at Hartland Point, Eskdalemuir and Lerwick to provide new data sets. We describe in detail how the systems were installed, their history, the electronics used to condition and digitize the signal, and how the data are processed and supplied in near real time to users. During more than a decade of measurements, we encountered several operational issues requiring mitigation and developed improvements as we gained experience of the systems.

#### 1 Introduction

During severe geomagnetic storms, large changes (of up to 10% of the total strength) in the Earth's magnetic field can occur. These rapid changes in the magnetic field induce an electric field in the conductive subsurface of the Earth, whose magnitude depends on the intensity and rate of the magnetic field change as well as the underlying geology of the local region. For example, a 1-in-100 year extreme geoelectric field calculated for the continental United States is modelled to be 24.3 V km<sup>-1</sup>, while the largest value that occurred during the March 1989 geomagnetic storm, the largest storm of the last century (Boteler, 2019) is suggested to be around 20 V km<sup>-1</sup> (Love et al., 2018).

Such events can cause serious damage to technological systems that make use of long conductors with earthing points at the ends. As the local geology influences the level of geoelectric field developed due to the change in geomagnetic field, these earthing points can have different electrical potentials, and current will flow preferentially through the conductor between them

if the electrical resistance is low enough. These are called Geomagnetically Induced Currents (GIC) and they can be significant. For example, Wik et al. (2008) list measured GIC ranging between 19A and 269 A at a particular transformer in Sweden during various geomagnetic events.

GIC can have a number of effect on high voltage power systems as even a relatively small GIC of tens of amperes can move a transformer into half-cycle saturation (Molinski, 2002). When in saturation, harmonics are generated from the biased hysteresis loop, which can cause instabilities in the power system such as tripping of protective systems and, in the worst case, heat damage to the transformer core. GIC flowing along high-pressure gas transmission pipelines can cause a voltage difference between the ground and the pipeline. Ingham et al. (2022) analysed the recorded voltages of cathodic protection systems on high pressure gas transmission pipelines in New Zealand during geomagnetic events. They found that in certain cases the protection systems were at risk of being pushed out of operational specification by GIC. In extreme cases, a risk of damage to equipment or even electrocution could occur.

Given these potential effects, it is important to have models which correctly predict the level of GIC in grounded technological systems. These models require an understanding of the underlying structure of the Earth and the surface impedance, often using geological survey data and mathematical modelling (Beggan et al., 2013; Bailey et al., 2018) or directly measured magneto-telluric (MT) impedances (Love et al., 2018; Campanyà et al., 2019; Malone-Leigh et al., 2024; Pratscher et al., 2024).

To validate models of GIC, a ground-truth measurement is required. Often this is a GIC measurement taken directly from the a transformer earthing point in the power grid (Bailey et al., 2018; Pratscher et al., 2024), but it is also valuable to have direct measurements of the geoelectric field available for comparison or where monitoring of current in the power or pipeline network is limited. This allows for comparison of models to a wide variety of geoelectric field conditions, during quiet and active geomagnetic conditions, enabling a better understanding of where there may be vulnerabilities in the grid. Long-term measurements of geoelectric field are also useful for understanding long-term trends in the geoelectric field, but such 55 installations are rare. Sites for the long-term measurement of geoelectric field to capture seismic events are present in Taiwan (Telesca et al., 2014) and Japan (Uyeda et al., 2000) although these are not co-located with magnetometers for close comparison between geoelectric and geomagnetic field changes. The longest running observations co-located with a site that also makes geomagnetic measurements were at Kakioka observatory in Japan (Fujii et al., 2015) recording from 1932 to 2021 (Nagamachi et al., 2023). Nagycenk observatory in Hungary (Ádám et al., 2009) has recorded since 1957. Other installations with geomagnetic and geoelectric recordings are at Boulder geomagnetic observatory, Colorado, United States (Blum et al., 2017) and Wittstock, Germany (Ritter, 2005). In the UK, there are records of some geoelectric field monitoring in the mid to late 19th century in London, but no long-term measurements until recently. In 2012, we began a programme to install geoelectric field monitoring sites at the three British Geological Survey (BGS) INTERMAGNET geomagnetic observatories in the UK.

- These sites have now been running for over a decade and have recently been upgraded to use new, more robust and reliable electronics. They provide data for studying the electric field and validating BGS geoelectric field models, which is also supplied in real time to the European Space Agency (ESA) space weather portal as part of the Geomagnetism Expert Service Centre (G-ESC).
- In this paper we give an outline of the development of the installed geoelectric field monitoring system over time, describe the components of the current system, discuss the operational challenges we have faced and summarize the data and scientific outputs of the system.

## 2 System description

## 2.1 Layout and Location

Geoelectric field monitoring systems are installed at the three UK INTERMAGNET standard observatories shown in Figure 1: Hartland (HAD), Eskdalemuir (ESK) and Lerwick (LER). Each of these sites was originally chosen for their large distance from sources of anthropogenic interference that could affect geomagnetic readings, such as electric railways. ESK is located in a rural valley in the Scottish Borders, HAD near the town of Hartland in Devon on the Bristol Channel and LER on a peninsula to the west of the town of Lerwick on the Shetland Mainland. These sites have long histories of geomagnetic recording, with absolute measurements dating back to 1908 at ESK, 1928 at LER and 1957 at HAD.

Figure 1 Location of UK Geomagnetic Observatories. Contains OS data © Crown copyright and database rights 2025 and NEXTMap Britain elevation data from Intermap Technologies.

- The geoelectric field monitoring system is made up of two pairs of LEMI-701 low-noise Cu-CuSO4 telluric electrodes arranged in two perpendicular arrays (Prystai and Pronenko, 2015). These electrodes provide a low-resistance path to ground in wet soil of 500Ω and a maximum self-potential difference of 0.06mV. The low noise, low contact resistance, high stability nature of the electrodes allows for a shorter line-length making them practical for observatories with limited space. Installing electrode runs more than 100m apart was impractical at the BGS observatories. The installation at LER is constrained by a nearby lake and the size of available land area and at HAD the electrode locations are limited by the size of the observatory site which is close to a small village which growing and beginning to encroach. The ESK electrode set-up is constrained by the topology of the terrain, being sited on a hill with other sensitive geophysical instrumentation and support infrastructure to the south, and boggy wet ground to the west.
- The electrodes were placed as far apart as possible within the limits of the local geography. At LER and HAD they are installed orthogonally to each other on the north-south and east-west cardinal directions. They are deployed in an L shape at LER and a T shape at HAD. However, during a recent resurvey of the ESK site it was discovered that the electrodes were not orthogonal, an error introduced during the original installation. The consequences and planned resolution to this are noted in the discussion

section. Layouts for the three observatories are shown in Figure 2 and the spacing, orientation and date of installation for each observatory is noted in Table 1.

50m 50m (b)

Figure 2: Location of electrodes at HAD (a), ESK (b) and LER (c). Basemap: (Esri, 2009), sources Esri, Maxar, Earthstar Geographics, and the GIS User Community.

| Observatory       | North-South | North-South | East-West   | East-West   | Date of      |
|-------------------|-------------|-------------|-------------|-------------|--------------|
|                   | Spacing (m) | Bearing (°) | Spacing (m) | Bearing (°) | installation |
| Hartland (LER)    | 69          | 179         | 76          | 267         | May 2013     |
| Eskdalemuir (ESK) | 66          | 192         | 86          | 268         | Nov 2012     |
| Lerwick (LER)     | 99          | 179         | 91          | 271         | Mar 2013     |

Table 1: Electrode spacing and date of initial system installation at each observatory

The electrodes are buried approximately 0.75 m deep as per the manufacturer's instructions and embedded in bentonite clay mixed with copper-sulfate salt to ensure a good connection to the ground. This reduces the effect of short-term temperature variation and provides a buffer against transient changes to the electrode-ground interface due to rain. These are emplaced as close as possible to the INTERMAGNET standard magnetometers (normally within 100m) at each site to ensure a consistent relationship between the geomagnetic and geoelectric field measurements.

The electrodes output wires are spliced to shielded coaxial cables. These are run through buried ducts to the interface hub and 115 the digitizer electronics. As these are in areas where livestock are present, burying the cables is essential to ensure they are not damaged by grazing. The cable shields are earthed at a common earth point at the electronics enclosure hub to reduce noise susceptibility in the long cables. This is the only earth point, as earthing at both ends can create ground loops.

After digitisation the data are transmitted back through a buried duct to a recording PC in the main recording hut. Power is supplied through the same duct at 15 V DC from the power supply in the recording hut.

#### 120 **2.2 Development History**

125

130

145

The system was initially installed on an experimental basis at all three observatories using a seismic digitizer with a preamplifier. This recorded at 100Hz and at a x10 scale and was operational from 2013-2015. However, such a high cadence was unnecessary and greater precision of the measurements could be achieved with greater amplification. During this period the HAD and ESK installations consistently produced data, however LER suffered from significant noise effects, as well as problems with large offsets away from zero. This limited the quality of the data returned from that site.

A second iteration of the system was installed at the end of 2015 at all three observatories, when the electronics were replaced by a system that included a x100 amplification channel as well as a x10 channel to allow both narrow-range, high-precision and wide-range, low-precision data to be simultaneously captured. This also introduced rudimentary lightning protection using diodes and a new ADAM-4017 industrial digitizer that was more compact, lower power and had existing integration with the BGS data recording software. A new set of LEMI-701 probes were installed at LER in 2017 to resolve the persistent data quality issues, rectifying the drifting and offsets that had previously been seen. This iteration of the system was in place between 2015 and 2021.

However, over time the diodes providing the lightning protection degraded and introduced increasingly large offset voltages into the system, causing the recorded data to drift and eventually reach the upper voltage limit of the digitizer. At the same time the probes that were installed in 2013 were beginning to degrade. The combination of these factors significantly reduced the reliability and data quality of the systems at all three observatories from mid-2018. A new iteration of the electronics was planned but the departure of key staff at BGS and the global COVID-19 pandemic delayed the refurbishment of the systems until 2021.

The third-generation system was installed between November 2021 and June 2022 at the three observatories, making use of improved amplification circuits to restore the systems to full operation. "Add-on" filtering circuits were added at ESK and HAD to reduce the effect of anthropogenic noise on the systems in 2022. LER was left without a filter board as the noise affecting it was within the 0-5Hz passband for the system. Additional filtering was assessed as unlikely to improve the data. The probes were replaced at ESK in 2021 and at HAD in 2022. The combination of the new probes and new electronics restored the data quality of the monitoring system and eliminated the electronic drift problem. However, lightning damage was a persistent issue at HAD resulting in multiple periods of downtime. The LER and ESK sites have been consistently operating and producing good scientific data since the upgrades.

The most recent (4<sup>th</sup>) generation of the system (described in the next section) consists of an integrated filter-amplifier stage combining the two boards of the third-generation system and reducing the size of the system. It has integrated and external lightning protection to address the issues with lightning damage and has additional design-for-maintenance features included to make repair easier. All three sites were upgraded to this standard during 2024. After upgrades or repairs the data were checked visually after a week to ensure periods of space weather activity at other operating stations, usually LER due to its consistent uptime, were reflected at the upgraded site. The dates of upgrades at each site are detailed in Table 2.

| Upgrade                       | ESK      | HAD      | LER      |
|-------------------------------|----------|----------|----------|
| Initial installation          | Nov 2012 | May 2013 | Mar 2013 |
| Second Generation Electronics | Sep 2015 | Nov 2015 | Oct 2015 |
| Third Generation Electronics  | Mar 2021 | Mar 2022 | Oct 2021 |
| Replacement Probes            | Jul 2021 | Jun 2022 | Jun 2017 |
| Add-on Filter Board           | Dec 2021 | Sep 2022 | N/A      |
| Fourth Generation Electronics | Nov 2024 | Aug 2024 | Nov 2024 |

Table 2: Upgrades and their installation dates for LER, ESK and HAD geoelectric monitoring systems

## 2.3 Digitization electronics

The electronics system consists of the electrode itself, a lightning protection system bonded to a safety earth, a pre-filter, an amplification stage, an analogue to digital converter (ADC), and a recording PC running the BGS' Simple Data Acquisition System (SDAS) software. The system block diagram is shown in Figure 3.

Figure 3: Geoelectric field recording electronics system block diagram. See text for further details

## 165 2.3.1 Lightning Protection

170

Early iterations of the system lacked comprehensive lightning protection, causing damage to the power system and the input stage amplifiers during lightning storms (see Figure 4). While a direct strike is difficult to protect against, measures were taken to harden the system against transients caused by earth potential rise and induced voltage from local strikes. The necessity of lightning protection was noted in Blum et al. (2017) and the construction of our lightning protection follows a similar template to that used at the USGS Boulder geomagnetic observatory in their geoelectric field observations, with some tailoring to suit our equipment.

Figure 4: Example of damage to an electronic chip from a lightning storm at HAD in 2022

The primary lightning protection for each channel is either an ABB Furse ESP15E (ABB Furse, 2024) or LEC Global DLP30 (LEC Global, 2012) surge arrestor bonded to an earth spike installed next to the system. These are fast-acting clamping circuits designed to keep the transient voltage below a set limit. Secondary lightning protection is integrated onto the amplifier printed circuit board (PCB) consisting of current limiting resistors and an Analog Devices AD4177 overvoltage protected (OVP) amplifier. This is capable of withstanding voltages 32V higher than the 12V line voltage, which is more than the maximum 39V worst case let-through voltage of the primary lightning protection (Analog Devices, 2018). The current limiting resistors ensure that the surge current is lower than the maximum allowable input current of the amplifier.

The trade-off to this protection is the addition of thermal noise due to the added inline resistance. However, as the bandwidth we are using is between 0 and 5 Hz, the actual amount of noise is very small. This can be calculated using Nyquist's formula for resistor thermal Root Mean Square (RMS) noise, given in Equation 1:

$$V_{RMS} = \sqrt{4k_b T R \Delta f} \tag{1}$$

where  $k_b$  is Boltzmann's constant, T is absolute temperature, R is resistance and  $\Delta f$  is the bandwidth. Given R of 10 k $\Omega$ , a T of 296.15°K and  $\Delta f$  of 5 Hz, the total calculated noise voltage is 28.6nV RMS per electrode, or 5.7uV RMS per channel at the x100 output after summing and amplification. This is significantly below the sensitivity of the ADC and can therefore be considered negligible.

190

Isolation is an important element of such systems, particularly when the power source is electrical mains. The mains earth can couple to the buried electrodes, as these are acting as earth connections in the circuit, to create ground loops. These enhance

noise pickup due to the large loop area, creating large offsets and posing a hazard during lightning events. As the addition of an extra earth can complicate the operation of the lightning protection it is not used. Traco TEC 2 1210 DC/DC converters with 1600V 60s isolation rating are used to isolate the input power from the pre-amplifier power and are themselves protected with DLP-30 lightning protection on the power inputs. Due to their vulnerability in the event of lightning strikes, the power systems are socketed rather than soldered to allow for easy replacement in the field.

## 2.3.2 Amplification stage

To provide good sensitivity across different ranges of input signals, an amplifier stage is used to scale the signal before digitization. The AD4177 input amplifier has a high input resistance of 4MΩ, minimizing the current in the input circuit to avoid damage to the electrode contact surface while still providing overvoltage protection and a low offset drift over temperature. Two amplification stages are used to provide a differential amplification of each channel by x10 and x100. This provides ±10 V km<sup>-1</sup> and ±1 V km<sup>-1</sup> ranges. For most situations the higher sensitivity, lower range x100 output is used. However, during a severe geomagnetic storm at higher-latitude observatories the x10 data from the secondary output can be used to capture the true magnitude of the storm when the primary output saturates. An example of this is shown in Figure 5, recorded during the October 2024 geomagnetic storm at ESK. The peak incident geoelectric field during the commencement of the storm exceeded 1 V km<sup>-1</sup> in both channels and both the positive and negative sides of the E/W channel. The x10 channel was used to determine the peak geoelectric field of 1.56 V km<sup>-1</sup> (i.e. 1,560 mV km<sup>-1</sup>).

195

Figure 5: Data recorded at ESK 15:00-16:00 UT 10-10-2024 - x10 channel, 10Hz

Zero drift, low noise LTC2057 amplifiers were used for the amplification chain to minimize the signal drift due to temperature variation and system aging. The maximum temperature drift on these amplifiers is  $0.015~\mu V$  °C<sup>-1</sup> with a maximum initial offset voltage of 4  $\mu V$  (Analog Devices, 2013). The noise level within the DC to 10Hz range is limited to 200 nV p-p. These characteristics mean the amp is quiet and self-calibrating, making it nearly immune to long term drift. This was selected to resolve issues of long-term amplifier drift that affected previous iterations of the conditioning electronics. The input ADA4177 amplifiers have a higher maximum offset and drift over time of 60  $\mu V$  and 1  $\mu V$  °C<sup>-1</sup> respectively, but this is an acceptable trade-off for the enhanced overvoltage protection they provide.

An offset correction circuit allows the system to be tuned during geomagnetically quiet periods to zero, eliminating any local earth self-potential difference. This minimises the effect of any static offsets between the electrodes, maximizing the positive and negative voltage range available to the digitizer during periods of geomagnetic activity. This offset is recorded by the digitizer on a secondary channel to allow for removal later if needed.

Due to the densely populated nature of the UK, anthropogenic interference is difficult to avoid. Two of the three observatories, LER and HAD are located near urban areas and ESK is close to areas of long-term forestry logging activity. Various human activities can cause spikes and steps, as well as power line noise and periodic interference. Therefore, filtering is required to reduce the worst of this unwanted noise. The amplifiers are pre-filtered using a three-pole Butterworth low-pass filter with a 5Hz passband. They are implemented on the PCB using an operational amplifier based active filter in Sallen-Key topology with unity gain. The effect of adding this pre-filter on anthropogenic interference is shown in Figure 6, Figure 7a and Figure 7b where an add-on filter was installed during a service visit on 6th December 2022 at HAD. The broad-spectrum interference was reduced significantly, particularly on the EW channel. The trade-off to the reduction in external interference is a series of features that likely stem from pickup of internal radiated signal from inside the electronics box itself. The add-on board had flying leads to the mainboard that, although configured as twisted pairs, increased the sensitivity of the system to locally radiated interference. These were preferable to the signal being washed out in anthropogenic interference, but it does highlight the trade-offs that need to be made when operating a system in a noisy environment. Integrating the filter board onto the main amplifier board in the latest iteration of the board has eliminated these internal electromagnetic interference bands (see Figure 7c and Figure 7d).

Figure 6: Effect of adding pre-filtering to reduce anthropogenic interference at HAD, x100 channel 10s data, from 00:00 UT 05-12-2022 to 00:00 UT 08-12-2022.

Figure 7: Power spectral density of data from HAD x100 10Hz EW channel (a) and NS channel (b) from 00:00 UT 05-12-2022 to 00:00 UT 08-12-2022. The add-on filter board was installed at just before 12:00 UT on 06-12-2022. Power spectral density from HAD x100 10Hz EW channel (c) and NS channel (d), 00:00 UT 05-12-2024 to 00:00 UT 08-12-2024 using 4<sup>th</sup> generation electronics with integrated pre-filter.

A second filter stage of the same design as the first is placed before the x10 tap and the x100 amplifier to remove any noise injected from the difference amplifier and the offset mixing.

The PCB is a four-layer construction with planes dedicated to ground and voltage rails to minimize current loop areas and the effect of electromagnetic interference.

# 2.3.3 Digitization and data collection

The outputs are digitized using an Advantech ADAM4017 digitizer with a respective sensitivity of 30 μV km<sup>-1</sup> and 300 μV km<sup>-1</sup> for the x100 and x10 channels. The digitizer collects at 10Hz and transmits the data back to the recording PC in the main recording hut using the RS485 protocol. The collection PC runs SDAS configured with the capture settings for the geoelectric-field system. This records and scales the data based on the distance between the electrodes to mV km<sup>-1</sup>, as well as recording ancillary information such as the current DC offsets. Data are saved to a file with data points timestamped using a clock

synchronised to a time server using the Network Time Protocol (NTP). This allows for data buffering if communications to the site are disrupted. The data are collected by servers at the BGS Office in Edinburgh every five minutes. Each system is battery backed with a 12V, 38Ah lead-acid battery which allows for approximately a day of recording during local total power loss. Data continuity is ensured during communications failure through local recording of the data to a ring buffer, which is read by the data servers in Edinburgh when communications are restored.

## 265 3 Operational challenges

Running these systems over time has revealed various operational challenges that need to be overcome to produce consistently high-quality scientific data. Anthropogenic interference, lightning, instrument drift and equipment failure all present different challenges to maintaining continuity of data and providing coverage during geomagnetically active periods.

# 3.1 Anthropogenic interference

Technological electrical activities invariably create noise in the ground through both broadcast electromagnetic interference and signals introduced into the ground through earthing. Different noise effects can be observed depending on the source. Electric railways can introduce a significant amount of electromagnetic noise pollution into the ground which can interfere with geoelectric field monitoring equipment (Ishikawa et al., 2007). None of the observatories are close to DC electric railways which are the most severe sources of geoelectric interference due to leakage current from the rails. The observatory sites were specifically chosen to avoid the effects of the railways on the magnetic data, as this was the reason for the closure of the Greenwich and Kew observatories in the late 19th and early 20th century, respectively. As such, no noise is expected from this source.

Steps can also occur in data and have been observed both at fixed installations and mobile MT studies conducted by our group. These have usually been in areas where large pieces of inductive machinery operate periodically and were noted in Ádám et al. (1986) where a step was correlated with mineworking activities. Takahashi et al. (2007) identified rectangular step-like signals in their recordings from the Boso Peninsula, using direction finding to attempt to isolate the source. It may be possible to use a similar technique after removing the offsets from our data in post-processing, however this has not yet been attempted for our installations. An example of this type of interference recorded at ESK is shown in Figure 8 where a step in both channels, negative in N/S and positive in E/W, commences at 09:45 UT and ends at 12:43 UT. The timing of this large step during early working hours, stopping around local lunch time, implies an artificial causation. The immediate area around the observatory is host to several forestry logging sites which are cleared as trees mature, and it is possible that interference is being detected from the forestry machinery being switched on and off, driven by locally earthed remote generators. This type of interference is difficult to eliminate at source as it is unaffected by the low pass filter due to the DC-step dynamic.

Figure 8: Data recorded on 11-12-2024 at ESK, x100 channel, 10s filtered data. Note the step between 0945 and 1243 UT.

305

310

Impulses in the data from transient surges can also be occasionally seen in the data. These can be natural signals (such as lightning) or can be created through similar mechanisms to steps, when inductive loads are powered on. These can be difficult to distinguish from natural variation, particularly where they occur within periods of geomagnetic activity.

295 Periodic interference can also be introduced into the system from sources such as corrosion protection systems. Ádám et al. (1986) noted that low frequency periodic interference was introduced into their measurements by the corrosion protection system from a nearby pipeline. This type of interference can cause long lasting, high amplitude noise depending on the proximity of the electrodes to the source.

Periodic interference has been detected at LER, where the system is sited close to the freshwater Loch of Trebister. By connecting the electrodes in different patterns we were able to isolate the strongest interference pickup to the southern electrode. This signal is complicated, consisting of two different phenomena. The first is a mix of low frequency interference tones which are present at all times. These can vary in power and frequency but are usually 

Figure 9: Power spectral density of LER NS 10Hz x100 data, 00:00 UT 09-09-2024 to 00:00 UT 11-09-2024 (a). Power spectral density slice for 23:59:10 UT 10-09-2024 (b). Peaks labelled: a) 0.5Hz low frequency interference term. b) 0.55Hz AM-like interference. c) 0.9Hz AM-like interference. d) 1.25Hz AM-like interference.

We have, so far, been unable to identify any obvious sources of this signal in the local area. Without being able to positively identify the source of the interference, relocation of the electrode pair away from the Loch during the next electrode replacement cycle is the likely next step in attempting to eliminate it. However, this relies on it being a very local source around the Loch, which may not be the case.

## 3.2 Natural Phenomena

Natural sources can also affect the measured electric field potential. The primary sources identified at our sites are tidal signals, lightning impulses, and signal and drift from rainfall and other precipitation.

### 3.2.1 Tidal Signals

Tidal signals are notable for introducing long-period diurnal signals in the local geoelectric field. This is due to build up of charge along the coastline from the motion of the tidal flow through the geomagnetic field (Osgood et al., 1970).

This tidal signal at HAD, which lies on the Bristol Channel can be easily seen in Figure 10 superimposed over the baseline geoelectric field. This area has one of the world's largest tidal ranges (over 7 meters) confined to a large channel, which induces a readily detectable signal. Peak-to-peak variations during October and November 2024 reached 25 mV km<sup>-1</sup> in both north-south and east-west channels during the spring tide on the 16<sup>th</sup> of November.

Figure 10: Geoelectric field recorded at HAD, 27-10-2024 to 20-11-2024, x100 channel, 10s filtered data. The periodic tidal signature is clear along with space weather activity on the 7<sup>th</sup> November.

Generally, the tidal signal is dominant at HAD throughout the year except during times of extremely strong geomagnetic activity such as the May 2024 storm, which can be seen in Figure 11. For the purposes of studying space weather at UK

geomagnetic latitudes, tides do not have a significant effect on the usefulness of the data, due to the long period and relatively modest amplitude compared to strong geomagnetic storm induced electric field. It is also straightforward to remove this signal using an appropriate band-pass filter if required.

Figure 11: Geoelectric field recorded at HAD, May 1-31, 2024, x100 channel, 10s filtered data. Large space weather induced variations of the geoelectric field occurred on the 10-12<sup>th</sup> May 2024.

## 3.2.2 Lightning

345

350

Lightning creates a strong electromagnetic field impulse as it strikes the ground and through the return stroke. There are three main lightning modes that we need to worry about when operating in a relatively lightning-prone area, such as the southwest of England:

- 1. Direct strike, where lightning directly hits the system assembly or the service cables.
- 2. Ground Potential Rise (GPR), where charge flows down from the thundercloud to the ground during the return stroke, creating a surge of current and a large difference in potential between nearby earthing points.
- 3. Electromagnetic induction, where the strike emits an electromagnetic transient that can couple onto long conductors.
- All electronics are installed in low-lying boxes surrounded by significantly taller structures and fed with buried service cables, making a direct strike on any part of the electronics system unlikely. However, GPR and induction are both serious problems for a geoelectric monitoring system. Sekioka et al. (2007) conducted simulations of GPR and line induction lightning strike modes and their effects on a low voltage DC system (in this case a domestic dwelling with surge protection). Their simulations indicated that a strike at 100m from a 35m long transmission line can create peak potential voltages of 3-15 kV depending on

the lightning current. The effect is increased the closer to the conductor or the earthing point the strike occurs. A strike that created a 3kV potential between the electrodes would destroy the input amplifier without adequate protection.

The 100m long lines provide a large antenna onto which lightning transients can couple. The electromagnetic pulse from lightning can induce a large transient voltage at the mid-point of the cable and conduct it to ground through both ends. This presents the highest chance of damage to the communications and power regulation electronics at the data acquisition enclosure. This risk can be reduced through use of lightning protection circuitry, shielded cables and isolation at either end of the comms and power channels, as this reduces the chance of harmful overvoltage passing into the more delicate acquisition electronics or damaging the power circuitry.

365

The GPR case is more complex in that a strike that occurs 100 m from the ground points creates a relatively small potential rise of less than one kilovolt. This would still be enough to damage the input systems. A more extreme case would be where the strike occurred 25 m from an electrode, where a potential of 5-25 kV would be seen on the near electrode, based on simulations from Sekioka et al. (2007), while the electrode a further 100 m from that might only see a few hundred volts or less, leading to a large electric potential between each channel, and between the near electrode channel and the system ground.

375 GPR is a particularly acute issue due to the system essentially having five independent grounds: the primary ground at the mains power point in the recording hut and the four telluric electrodes. This can create a complex series of potentials during a lightning strike depending on which electrode is closest to the strike point. The electrode furthest from the strike might only see a few hundred volts, leading to a very large potential difference between the channels. In our systems, the electrodes have T or L shaped assemblies, and a strike at one electrode would generate large potentials between any two other electrodes, with 380 the amplitude varying depending on the location of the strike. The resulting potentials can cause breakdown in the input amplifiers between the channels, or between a channel and the power rails. The resulting fault current can damage or destroy the amplifiers or the power supply. Mitigating against this requires earthed lightning protection circuitry on each electrode channel at the electronics box. This allows a path for the fault current to discharge safely. Most commercially available lightning protection circuits specify a let-through voltage which is the maximum value the lightning surge will be clamped to. 385 A buffer amplifier that is capable of withstanding this let-through voltage should be specified, with an appropriate currentlimiting resistor specified to match the maximum input current of the chosen buffer amplifier. Appropriate isolation of power systems should also be considered to avoid introducing additional current paths that may reduce the effectiveness of the lightning protection.

Beyond the damage that lightning can do to a system, the impulses created by lightning are visible in data collected during storms. If the lightning protection system is active, then the input amplifier will be saturated. Figure 12 shows the lightning impulses during the evening of 5<sup>th</sup> August 2024, when a significant lightning storm was reported at HAD.

Figure 12: Lightning impulses captured during a storm at HAD 05-08-2024, 1 second data

## 3.2.3 Precipitation

An inescapable aspect of operating in the United Kingdom are periods of significant precipitation. When the soil becomes inundated with water the conductivity of the ground can change. In areas where one electrode becomes waterlogged, this can cause a large change in the measured geoelectric field which appears as a large drift over an hours-scale timeframe followed by a slow return to the original level over several days as the ground dries out. This can be seen in Figure 13 where two periods of heavy precipitation over the new year between 2024 and 2025 fell during a period of geomagnetic storm activity. The rainfall and subsequent waterlogging of the ground affected the north-south pair, which is emplaced in boggy terrain, causing the output to drift. This returned to a baseline level around zero over the next 48 hours of dry weather. The east-west electrode pair was not affected as these are further up the hill in a spot with better drainage. Fujii et al. (2015) encountered this issue at the Kakioka, Japan observatory in their observations from 2000-2011 where the baseline would drift due to rainfall. Our electrodes are buried at least 0.75 m below the surface to try to mitigate this, but in certain areas a deeper installation may be required. However, this isn't necessarily a solution to all sites, as at Kakioka the electrodes were buried 3 m deep and still experienced instability due to precipitation. In this case relocation of the electrode may need to be considered.

Figure 13: Example of precipitation-induced drift in the NS channel at ESK, 1 second filtered data

# 4 Outputs: Research and Applied

## 4.1 Geoelectric field modelling

One of the motivations for making these long-term measurements is for validating our electric field models. Beggan et al. (2021) used the measured electric field data to assess the performance of a thin-sheet electric field model during three storms between 2013 and 2018. This analysis provided important information about the limitations of the model, giving a better understanding of the uncertainties both in this model and the GIC model that relies upon it. Work is underway to produce an improved model of the geoelectric field in the UK, based on a magneto-telluric measurement campaign. These long-term measurements of the electric field will play an important role in validating that model in the future.

## 4.2 Geoelectric tides

Investigations to find several distinct tidal periods in the geoelectric field data were carried by Baillie (2020). The data were cleaned and processed first by creating one-second values from 10 Hz data using a median filter. A minimum of five non-NaN values out of nine were required to compute the median, otherwise the one-second value was set to NaN. The one-second data were passed through an adaptive Hampel-filter with an 11-point window to further clean the data. The final one-minute values were computed using a 61-point cosine filter centred on the minute mark, giving the off-centre values lower weight and tapering to zero at the edges.

Superposed Epoch Analysis (SEA) revealed the dominant lunar (M<sub>2</sub>) and solar (S<sub>2</sub>) semi-diurnal signals at all three sites. Fourier and Lomb-Scargle analyses unveiled further harmonics of tides of both lunar and solar origins including O<sub>1</sub> and S<sub>1</sub>. The size of these tidal harmonics was found to vary greatly between sites due to a number of factors, for example, distance from the coast, geology and local precipitation patterns. This work demonstrated that measurements from HAD correlate well with the rate of change of tide height at the closest tidal gauge station, Ilfracombe, confirming the hypothesis that the tidal flow in the Bristol channel is the main driver for the induced geoelectric field measured at Hartland observatory.

The picture is rather more complicated at LER due to the intricate interaction of tidal flow pattern between the Atlantic to the west of Shetland and the North-Sea, but most of the power is in the dominant motion induced, M<sub>2</sub> tidal signal. Geoelectric field data from ESK, the furthest site inland (70km from the coast), is noisier and the correlation with tide gauge data is weaker, though still present. Even so, the M<sub>2</sub> tidal signal was detected far inland, and had a greater magnitude in the east-west geoelectric field than north-south, which is explained by the manner in which tides travel around the Britain in a clockwise direction.

#### 440 4.3 Making the data available

The data are collected and stored within the BGS system at a sampling frequency of 10 Hz, which are available on request. This provides a large volume of data for our studies but is unwieldy for regular delivery over the internet. A filtered 1 Hz dataset is created using an 11-point cosine filter, centred on each second. This 1 Hz dataset is then filtered again (with an 11-point cosine filter) to provide a data with a 10 second sampling. There is currently no other continuous quality control, so steps, spikes, drifts and other noise remain in these data.

The 10 second data are delivered to the ESA Space Weather Service Network (https://swe.ssa.esa.int) and are available for plotting or download for any registered user. Plots of the 1 second data are also available via the BGS Geomagnetism website, and a new web service is currently in development to provide both filtered datasets.

#### 450 5 Discussion

445

455

430

The primary difficulty for our measurements was dealing with the various operational challenges. Of these issues, lightning presented the most common and frustrating threat. Due to the multiple points of grounding throughout the system, careful attention must be paid to the design of the lightning protection system, particularly the interaction between the sensor earthing and the safety earthing. Multiple iterations of the protection architecture were developed before identifying a scheme that worked well enough to resist the lightning events that occur regularly at HAD while not disrupting the operation of the system by creating large ground loops that pick up noise and interference. It should be noted that the LER installation has been running since inception without external lightning protection and has only required repair due to a surge on one occasion. We were

unable to definitively identify this as a lightning strike, although the damage was very similar. In contrast, HAD and ESK have both experienced multiple failures over the years due to lightning, prior to the most recent and more effective protection system. For systems that are in areas with limited lightning incidence, it may be more cost-effective to accept the occasional repair of the system, as the external lightning protection systems can be expensive to install.

Anthropogenic noise is the next-most disruptive issue when attempting to measure the geoelectric field. In the United Kingdom, the density of inhabitation means that very few locations can be sited far enough away from human activity to give a completely clean measurement of the ground. This means that additional filtering is required to reduce the impact of interference on the data quality; in our case an additional 5Hz cutoff, 3 pole Butterworth response analogue pre-filter. With the filtering in-place we were still able to capture valuable data in areas close to human activity. As the geomagnetic observatories were deliberately sited to avoid the influence of DC electric trains, we have, fortunately, not had to deal with that issue but it is something that must be considered in countries with extensive DC rail networks when siting a permanent geoelectric field monitor.

We noted in the section 2.1 Layout and Location that we had identified a layout error in the ESK electric field system. During a recent resurvey of the probe locations, prompted by difficulties in finding the emplacements during the 2021 probe replacement, we discovered that the south electrode had been installed at a bearing of approximately 192° from the north electrode. The east-west electrode bearing is 268°, which is within the survey margin of error to the ideal 270° east-west bearing. With a misalignment of 12°, approximately 21% of the signal magnitude of the east-west component will be coupled into the north-south signal, while the magnitude of the north-south signal will be reduced by approximately 2%. Due to the distance from the sea, we believe this will have a minimal impact on the geoelectric tidal signals we have analysed. However, it will have a greater effect on storm responses. The effect of the misalignment will depend on the relative potentials of the electric fields in each direction at any given time. A large north-south signal that is paired with a weak east-west signal will have a lower relative amount of error when compared to a large east-west signal paired with a weak north-south signal. This adds an unwelcome uncertainty to the gathered data, so we intend to run a series of tests and adjustments of the system in future.

A further resurvey with a more accurate differential GPS is planned to confirm the exact locations of the probes, as the handheld GPS has an error of 3 m, translating to a ±5° angle error. Once this is confirmed we plan to run a LEMI portable magneto-telluric system alongside the ESK system, with the east, west and north probes co-located as closely as practically possible, and the south probe in an ideal spot 100 m and at 180° bearing to the north probe. This will allow us to compare the two systems and the derived magneto-telluric tensor to determine the actual effect of the misalignment on the data, as the local geology is complicated. As ESK sits on a well-known conductivity anomaly at depth (Jain and Wilson, 1967; Wang et al., 2025), the effect of the cross-coupling could be magnified or reduced by this effect. Once this study has been completed, we

will move the south electrode to the new position. This has the added benefit of moving the electrode out of the boggy ground that it is currently sited in and into a slightly drier location.

## **6 Conclusions**

We have now been measuring the geoelectric field at our three observatories for over a decade. Our experience has demonstrated that, while measuring the geoelectric field can be challenging due to the difficult measurement environment, it is worthwhile to do so for the scientific insights gained mentioned in previous sections. We have detailed the various challenges encountered over this period with both natural and anthropogenic factors and, where possible, how we mitigated against them.

With the lessons learned over that time we now have a more robust and reliable experimental setup with which we intend to capture the geoelectric field in the long term. We have described in detail the deployment of our systems, the engineering justification for our decisions, and the upgrades that the systems have gone through to bring them to their present state. While some custom electronics were produced to integrate the equipment into the standard BGS data collection apparatus, this consists of relatively simple amplifier/filter conditioning electronics married to commercial off-the-shelf digitization equipment. With the availability of commercial recording equipment tailored to geomagnetic observatories, the installation cost for an entirely new system is modest and we hope that this encourages other institutions to consider installing long-term monitoring stations of their own.

The data from the observatories has proven useful for understanding space weather impacts and the natural variations in the geoelectric field caused by tidal motions. We have made the data available to the research community through the ESA Space Weather Service Network and the BGS website.

## **Author Contribution**

515

RL refurbished the present system, designed the updated electronics boards and lightning protection, wrote the technical description of the system for this paper and contributed to the introduction, operational challenges, discussion and conclusions sections.

GR has been involved in discussions on the operation of the system since the original installation of the systems. She also contributed to the introduction, outputs, discussion and conclusion sections of the paper, as well as making all the plots of electric field data.

OB wrote a summary of her master's thesis on geoelectric tides in the research outputs section and contributed to the introduction, discussion and conclusion sections.

#### **Competing Interests**

The authors declare that they have no conflict of interest.

## Acknowledgements

The initial installation and early upgrades to the system were carried out by Tom Shanahan and Tony Swan without who these measurements would never have been started. We are also very grateful for the support of the BGS engineers, observatory staff and colleagues at the Met Office who have been instrumental in keeping these systems running. We would also like to thank Kristina Rossavik and John Spritzer from USGS for their advice and valuable insights on lightning protection solutions for geoelectric field measurements. We would like to thank Harry Harrison for his insights into lightning protection from his experiences with electro resistive tomography work.

530

This work was supported by the British Geological Survey via NERC national capability. The original project was set up under the BGS OF7 opportunities fund in 2013.

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
