# Peer review of "Long term monitoring of the geoelectric field in the UK – 2012-2024"

_EGUsphere, 2025_

## Referee Comment (RC1)

Long term monitoring of the geoelectric field in the UK – 2012-2024

Robert Lyon1, Gemma S. Richardson1, Orsi Baillie1
1British Geological Survey, Research Ave South, Riccarton, EH14 4AP Edinburgh, UK
Correspondence to: Robert Lyon (rlyon@bgs.ac.uk)

**Reviewer Report**

**General Comment:**

Overall, the paper presents the site selection, design, installation, and ongoing development of geoelectric observation systems installed in 2012–2013 at three remote locations: Hartland Point, Eskdalemuir, and Lerwick. These sites were co-located with INTERMAGNET observatories and chosen to be as far as possible from anthropogenic noise sources. The development of the systems was primarily driven by the need to adapt to changes observed in the data series, to address typical failures, and to mitigate challenges arising from local anthropogenic influences.

The paper provides sufficient detail on the problems encountered and the solutions implemented, thereby demonstrating the practical difficulties of operating a permanent geoelectric monitoring system with high accuracy. It discusses aspects such as digitization, site-specific filtering of the data, and the real-time transmission of processed datasets to users and to the ESA Space Weather Portal.

In my opinion, the work described in the paper constitutes a valuable contribution to our understanding of geomagnetic induction and the coupling of space weather effects into the Earth system via induction processes. The resulting dataset fills a significant gap in the field. The team has done a thorough and careful job to extract the highest quality, low-noise, and drift-free geoelectric time series from the potential differences measured between electrode pairs at the three sites. They have found appropriate responses to the challenges encountered and have produced a dataset that will be instrumental in improving the modeling of geomagnetically induced currents (GICs) during space weather events.

**Section-by-Section Review:**

**Chapter 1** provides a clear overview of the motivation and scientific background for installing permanent geoelectric observation stations. The authors emphasize the importance of long-term monitoring of the geomagnetic field, both for capturing long-term trends and short-term variations. The chapter also includes a brief list of the very few existing permanent geoelectric observation sites worldwide, underlining the uniqueness and value of the presented work.

**Section 2.1** introduces the environmental and scientific considerations that guided site selection and installation. The practical choices regarding electrode geometry and materials are well justified, especially given the spatial limitations at each observation site.

**Sections 2.2 to 2.3.3** provide an authentic and thorough account of the key considerations in designing and implementing geoelectric field measurements. The paper outlines the challenges encountered during installation and operation, and how these were anticipated or resolved over the

course of ten years of operational experience. The authors describe the digitization process, potential sources of offset, and the iterative development of effective lightning protection systems, along with advances in multistage amplification techniques. Site-specific anthropogenic noise filtering is shown to be a critical aspect of geoelectric monitoring, requiring carefully considered trade-offs. These decisions are well illustrated with practical examples from the field.

**Chapter 3** draws attention to the typical challenges of detecting natural geoelectric signals, many of which are familiar from our own experience at the Széchenyi István Geophysical Observatory (SzIGO). These include various anthropogenic noise sources, instrumental drift, and leakage currents — such as those originating from DC railways or the grounding systems of nearby buildings. Step-like disturbances, which are notoriously difficult to filter out automatically, are also discussed. The authors highlight how periodic interferences are often tied to nearby local sources, varying by site, and not always easy to identify. At the LER site, they successfully investigated and identified most interference sources and applied appropriate filtering techniques.

Of particular interest is the observation of strong tidal signals at the HAD station — a phenomenon we do not observe at inland stations such as NCK. The paper also covers the signatures of distant lightning strikes and the effect of precipitation-induced drift in telluric field measurements. Section 3.2.3 clearly demonstrates how precipitation can influence geoelectric measurements if electrodes are not buried deep enough. At the Nagycenk station, for example, 1 m² lead plate electrodes are buried more than 2 meters below the surface. Despite this, some seasonal variation can still be observed, although the electrodes are installed on a hilltop.

**Chapter 4** provides an overview of the scientific applications and research directions enabled by this unique and valuable time series of permanent telluric observations. The data's accessibility via the ESA Space Weather Portal and the BGS Geomagnetism Portal further enhances its value for the wider research community.

**Comments:**

The HUN-REN Research Institute of Earth Physics and Space Science (and its predecessor institutions: MTA GGKI, MTA GGI MTA CSFK GGI, ELKH GGI) has operated a permanent telluric observation system at the Széchenyi István Geophysical Observatory in Nagycenk, Hungary, since 1956. This was the first geophysical measurement system established at the observatory, later supplemented by geomagnetic observations in 1957. The original electrode spacing was 500 meters in both the north–south and east–west directions, determined to match the sensitivity of the galvanometers available at the time. However, the system often operated in saturation during at least moderately disturbed intervals due to the lack of automatic range switching.

A new telluric monitoring system, developed by the Space Research Laboratory of the Budapest University of Technology and Economics, is currently being tested at the Nagycenk observatory. It operates with a 40 Hz sampling rate, filtered to an effective 10 Hz, and features dual-range recording: when the higher-resolution (narrow-range) channel saturates, the system automatically switches to a lower-resolution (extended-range) mode. This new system has been in operation with

a 250-meter electrode spacing since 2023. Similar solution to the same problem, as it is introduced in the paper under review in 2.3.2 subsection.

**Review based on the provided aspects:**

**1. Relevance to GI**

Yes, the paper addresses scientifically relevant questions within the scope of *Geoscientific Instrumentation, Methods and Data Systems (GI)*. It focuses on long-term geoelectric monitoring, hardware development, and operational challenges—central to GI's objectives.

**2. Novelty**

The paper presents several novel aspects:

- Over a decade of operational data from co-located geomagnetic and geoelectric sensors in the UK, previously non-existent.

- Iterative hardware developments specifically designed for geoelectric monitoring.

- Practical field lessons in lightning protection and anthropogenic noise mitigation.

- Insights into natural geoelectric signatures such as tidal effects and their site-specific variation.

**3. Conclusions**

Substantial conclusions are reached, particularly regarding:

- The feasibility and value of long-term monitoring.

- Engineering trade-offs for sensor robustness.

- Data utility for both space weather and Earth system science.

**4. Methods and Assumptions**

Yes. The technical descriptions (e.g., amplifier noise, ADC specs, filtering strategies, lightning modeling) are thorough. Assumptions—such as regarding signal frequency bands or tidal coupling —are realistic and explicitly discussed.

**5. Support for Interpretations**

Yes, the presented results are sufficient to support interpretations. The use of spectral density plots, case studies (e.g., May 2024 storm, tidal signals at HAD), and historical comparisons reinforce the validity of conclusions.

**6. Reproducibility**

Yes. The system's design, installation procedures, electronics schematics (described in text), and filtering/data acquisition pipelines are all detailed enough for reproduction, with some site-specific exceptions (e.g., terrain constraints).

**7. Credit to Related Work**

Yes. The authors cite key foundational and recent works in the field, such as Boteler (2019), Love et al. (2018), and regional studies (e.g., Ádám et al., 2009; Blum et al., 2017). They clearly distinguish their original contributions, especially hardware/system design and long-term UK data generation.

**8. Title Appropriateness**

Yes, the title accurately and succinctly reflects the content.

**9. Abstract Clarity**

Yes. The abstract is concise and informative. It defines the scope, significance, instrumentation, and goals clearly.

**10. Presentation Quality**

The structure is logical: introduction, system description, challenges, outputs, discussion, and conclusions. Figures are well-integrated and informative. The evolution of the system over time is clearly tracked.

**11. Language**

Generally fluent and precise.

**12. Mathematical Clarity**

Yes. Formulae (e.g., thermal noise via Nyquist) are used correctly and well contextualized. Units are consistent and appropriate throughout.

**13. Content Streamlining**

Minor suggestion: Figures 6–11 are crucial, but the narrative could benefit from grouping the spectral analysis figures (Figures 7–8, 10–11) into composite panels for compactness. Just a recommendation.

**14. References**

Yes. References are numerous, up-to-date, and include most key papers in the domain. The authors appropriately cite both foundational and niche studies relevant to their work.

**15. Supplementary Material**

There is no mention of separate supplementary materials. However, the description of data availability (e.g., via ESA space weather portal) is sufficient for transparency. Providing links to design schematics or PCB layouts in supplemental material might further enhance reproducibility.
* * *
**Recommendation**

I recommend **minor revisions** before acceptance. The paper is an excellent and much-needed contribution to the field of geoelectric monitoring and instrumentation. Its long-term perspective, detailed hardware development, and integration into broader space weather networks are especially commendable.

**1. Clarifications or Expansions Recommended**

**1.1. Cross-Talk Due to Misalignment at ESK**

- **Issue**: The paper mentions a ~12° misalignment at ESK leading to ~20% cross-coupling (Section 5), but stops short of quantifying its impact on the long-term dataset.

- **Suggestion**: Include a rough estimate or preliminary analysis of how much this misalignment might bias tidal studies or storm response data.

**1.2. Filter Design Parameters**

- **Issue**: The filtering is critical in dealing with anthropogenic noise, but only the **type (low-pass, 5 Hz)** is mentioned.

- **Suggestion**: Please add the **filter order** (e.g., Butterworth, Bessel?) and analog vs digital implementation details. This improves reproducibility and transparency for others aiming to replicate the design.

**2. Small Technical or Editorial Corrections**

**Typographical / Stylistic Issues**

- **"In areas where one electrode becomes waterlogged this can cause..."**
  → insert a comma after "waterlogged".

- A few places use **"this required mitigation"** repeatedly. Vary phrasing or tighten wording (e.g., *"mitigated by…"*).

- In line 428, you mention **O1** — could this be a typo? Should it perhaps be **M1?**
- In kine 471: Locationthat

**3. Possibly Missing Aspects Worth Mentioning**

**3.1. Instrument Calibration and Inter-site Consistency**

- **Missing**: The paper does not state how consistency between sites was validated after each hardware upgrade.

- **Suggestion**: Add a sentence about whether **cross-calibration or inter-site comparison** (e.g., response to shared storms) was done to ensure uniformity in measurements.

**3.2. Power Backup or Data Gaps**

- **Missing**: No mention of **backup power, data loss handling**, or system uptime statistics.

- **Suggestion**: One sentence on whether power outages caused downtime or how data continuity is ensured (e.g., battery backup, local buffering) would close this small gap.

---

## Author Response (AR1)

**Author's response to reviewer comments – Long term monitoring of the geoelectric field in the UK 2012-2024**

Robert Lyon1, Gemma S. Richardson1, Orsi Baillie1

1British Geological Survey, Research Ave South, Riccarton, EH14 4AP Edinburgh, UK

Reviewer comments noted in italics, response in normal text.

**Reviewer RC1 - István Lemperger**

Minor suggestion: Figures 6–11 are crucial, but the narrative could benefit from grouping the spectral analysis figures (Figures 7–8, 10–11) into composite panels for compactness. Just a recommendation.

We agree with this assessment and have condensed these down to composite figures 7 and 9.

Modified individual panels to the combined panels on lines 245 and 315

There is no mention of separate supplementary materials. However, the description of data availability (e.g., via ESA space weather portal) is sufficient for transparency. Providing links to design schematics or PCB layouts in supplemental material might further enhance reproducibility.

It is our intention to upload these schematics as open hardware, however we are in discussion with BGS' intellectual property team on the proper way of making these schematics available in line with institute policy. We currently plan to host the schematics on an open-access BGS or NERC platform and link to them from the electric field product website.

1.1. Cross-Talk Due to Misalignment at ESK

Issue: The paper mentions a  $\sim$ 12° misalignment at ESK leading to  $\sim$ 20% cross-coupling (Section 5), but stops short of quantifying its impact on the long-term dataset.

Suggestion: Include a rough estimate or preliminary analysis of how much this misalignment might bias tidal studies or storm response data.

We agree that a small amount of initial analysis would be useful, although prior to more detailed investigations we don't want to publish anything that may not be substantiated in the future.

Lines 476-481: We have included some additional, preliminary analysis on how we think this might affect the data and why. This helped clarify to us that magnetic vector direction of the storm would affect how severe the distortion of the data would be.

1.2. Filter Design Parameters

Issue: The filtering is critical in dealing with anthropogenic noise, but only the type (low-pass, 5 Hz) is mentioned.

Suggestion: Please add the filter order (e.g., Butterworth, Bessel?) and analog vs digital implementation details. This improves reproducibility and transparency for others aiming to replicate the design.

Agreed, this adds to the reproducibility of the design.

Line 229-231: We have added the requested information around the filter response, order and analogue electronics implementation.

2. Small Technical or Editorial Corrections Typographical / Stylistic Issues

"In areas where one electrode becomes waterlogged this can cause..."  $\rightarrow$  insert a comma after "waterlogged".

Line 398: Comma added

A few places use "this required mitigation" repeatedly. Vary phrasing or tighten wording (e.g., "mitigated by...").

Line 365, Line 466: Agreed. We have replaced several instances of mitigated (or similar) with suitable synonyms or more precise terminology.

In line 428, you mention O1 — could this be a typo? Should it perhaps be M1?

This is the correct notation.  $O_1$  is used to define the Lunar diurnal tide caused by the Moon's gravitational pull, with a period of 25.82 hours.

In line 472: Locationthat

**Space added**

- 3.1. Instrument Calibration and Inter-site Consistency
  - Missing: The paper does not state how consistency between sites was validated after each hardware upgrade.
  - Suggestion: Add a sentence about whether cross-calibration or inter-site comparison (e.g., response to shared storms) was done to ensure uniformity in measurements.

We agree this would be useful. Our inter-site comparison process isn't complicated but does confirm the site is working and responding to space weather so we should include it.

Line 153: Added a short description of our process of comparing upgraded sites to working existing sites.

3.2. Power Backup or Data Gaps

Missing: No mention of backup power, data loss handling, or system uptime statistics.

Suggestion: One sentence on whether power outages caused downtime or how data continuity is ensured (e.g., battery backup, local buffering) would close this small gap.

Agreed, this is an easily missed step in ensuring data continuity so should be included.

Line 260-264: Added additional information about the power and data backup systems.

**Reviewer RC3 - Anonymous**

In the Introduction: please, use either "GIC" or "GICs". You first use "GIC" abbreviation: "Geomagnetically Induced Currents (GIC)". You can continue using "GIC" hereinafter.

Line 31-50: Agreed, amended GICs to GIC for consistency

Instead of the "ESA Space Weather portal", you can use the official name "ESA Space Weather Service Network" and provide the link: <a href="https://swe.ssa.esa.int">https://swe.ssa.esa.int</a>

Thank you for the correction. The link to the ESA service is also worth including.

Line 447: Corrected name of ESA Space Weather Service Network and added the link to the service in the text body.